# Math and language gender stereotypes: Age and gender differences in implicit biases and explicit beliefs

**Heidi A. Vuletich**[1]*, **Beth Kurtz-Costes**[2], **Erin Cooley**[3], **B. Keith Payne**[2]

**1** Department of Psychological and Brain Sciences, Indiana University Bloomington, Bloomington, Indiana, United States of America, **2** Department of Psychology and Neuroscience, University of North Carolina at Chapel Hill, Chapel Hill, North Carolina, United States of America, **3** Department of Psychological and Brain Sciences, Colgate University, Hamilton, New York, United States of America

* hvuleti@iu.edu

**Data Availability Statement:** All relevant data are uploaded to the Open Science Framework database and publicly accessible via the following URL: https://osf.io/fv5h8/.

## Abstract

In a cross-sectional study of youth ages 8–15, we examined implicit and explicit gender stereotypes regarding math and language abilities. We investigated how implicit and explicit stereotypes differ across age and gender groups and whether they are consistent with cultural stereotypes. Participants ($N = 270$) completed the Affect Misattribution Procedure (AMP) and a survey of explicit beliefs. Across all ages, boys showed neither math nor language implicit gender biases, whereas girls implicitly favored girls in both domains. These findings are counter to cultural stereotypes, which favor boys in math. On the explicit measure, both boys' and girls' primary tendency was to favor girls in math and language ability, with the exception of elementary school boys, who rated genders equally. We conclude that objective gender differences in academic success guide differences in children's explicit reports and implicit biases.

## Introduction

Children's perceptions of gender differences in cognitive abilities are important because they may lead boys and girls to develop different interests and different areas of achievement [1–3]. For example, stereotypic perceptions of academic abilities have been implicated in gender differences in course selections and career trajectories, contributing to the underrepresentation of women in science, technology, engineering and math (STEM) [4,5]. Much of the research devoted to this topic has examined children's self-reported *explicit beliefs* about gender differences in abilities. As summarized below, explicit beliefs may differ in systematic ways from *implicit gender biases*, which are automatically activated associations to gender categories. In this study, we examined children's implicit biases and their explicit beliefs regarding gender differences in math and language abilities.

Math is directly relevant to students' interest in and preparation for STEM careers, but biases favoring girls in language could also contribute to STEM disparities by disproportionately attracting girls and deterring boys from the humanities [6]. To identify if, and when,

**Funding:** This research was supported by a grant from the National Institute of Child Health and Human Development (https://www.nichd.nih.gov/) #1R03HD072025-01A1 awarded to BKC and KP. The first author was supported by a National Science Foundation Graduate Fellowship and the Paul and Daisy Soros Fellowships for New Americans. The funders had no role in study design, data collection and analysis, decision to publish, or preparation of the manuscript.

**Competing interests:** The authors have declared that no competing interests exist.

changes in explicit beliefs and implicit biases are taking place, we examined age differences in these phenomena, using a sample that ranged in age from 8 to 15 years. We also examined relations between youth's implicit biases and their explicit beliefs.

## Possible influences on children's perceptions of gender differences in academic abilities

According to social identity theory, identifying with a social category leads to in-group preferences because doing so protects people's self-esteem [7]. Gender is one of the first social categories to develop, and children as young as 3 to 5 years old express preferences for their own gender [8,9]. Same-gender preferences typically result in more generalized positive evaluations of one's gender group, particularly among children [10]. For these reasons, one would predict that children's explicit reports of competence in academic domains would be biased by their gender, resulting in in-group preferences. Other factors that may influence children's perceptions of academic abilities, particularly among older children, are the need to be fair [11], the development of more accurate evaluations [12], and the need to present themselves in socially desirable ways [13].

As children age, they also become aware of cultural stereotypes that contradict their positive evaluations of the in-group. Two traditional gender stereotypes are that boys are more talented than girls are in math, whereas girls are more talented than boys in language domains [14]. Sociocultural views of knowledge construction suggest that children's awareness of these kinds of cultural stereotypes increases with maturation [15], and empirical studies have shown support for this idea [16–18]. Therefore, as children grow older and have more experiences in which stereotypes are discussed or endorsed by others, these stereotypes may become a more salient source informing children's own attitudes. In particular, when children encounter negative stereotypes about their social group, those stereotypes may either temper their in-group favoritism, yielding more egalitarian attitudes, or guide their attitudes such that they endorse the negative stereotypes about their group. Research suggests that the way children respond often depends on where their group is positioned within the larger social hierarchy [19]. Children belonging to groups of high status are more likely to endorse negative stereotypes about their in-group, presumably because they do not have a strong need to self-enhance. Children belonging to low-status groups are more resistant to endorsing negative stereotypes and either downplay them or opt for egalitarian views instead. For instance, in two studies, boys and Whites endorsed traditional gender and race stereotypes regardless of whether those stereotypes favored their in-group, but girls and Black children were more selective, endorsing stereotypes that favored their in-group while denying stereotypes that portrayed their in-group negatively [18,19].

Another factor influencing children and adolescents' beliefs is their personal experience. Although in most countries worldwide men are more likely than women to pursue careers in STEM domains [20], girls tend to receive higher grades than boys throughout primary and secondary school across academic subjects [21,22]. Particularly as youth enter middle school, where achievement becomes more salient because of academic tracking and public posting of honor rolls, they are likely to become aware of differences in academic performance favoring girls. Indeed, using a photo-identification task in which youth were asked to pair photos with verbal descriptions of individuals, middle school youth were more likely to choose photos of girls than of boys for depictions of high-achieving youth [23].

Girls' relative superiority over boys in academic settings is robust, as reflected by school grades, high school and college completion rates, and teachers' ratings of school engagement [21,24,25]. Therefore, although gender differences persist in career choices, and stereotypes

are still widely endorsed that favor boys in math, youths' awareness of gender differences in school success might lead to implicit biases and/or explicit reports that are either egalitarian or that favor girls across domains. For instance, one research study with children ages 4–10 found that, with age, boys increasingly endorse personal beliefs about girls' academic superiority and also beliefs that adults see girls as academically superior [26].

## Explicit beliefs regarding gender differences in academic abilities

As outlined in the previous section, several factors potentially influence children's perceptions of gender differences in academic abilities. Unfortunately, research studies investigating children's beliefs through explicit reports have not demonstrated consistent findings in support of any one explanation. Regarding math ability, several studies have shown that young boys (approximately ages 6–11) show in-group preferences [27,28]. In two different studies, Italian boys in first grade were more likely to point to a picture of a boy, rather than a girl, when asked who is especially good at math [28,29]. Italian boys in third and fifth grade were also more likely to say that boys, rather than girls, were better at math [30]. In U.S., German, and French samples, boys in fourth grade rated boys as more gifted than girls in math [31–33]. Singaporean boys in first, third, and fifth grade were more likely to point to a picture of a boy, rather than a girl, when asked who liked math more [27]. These results are consistent with the idea that in-group preferences dominate in early and middle childhood. However, this explanation does not hold for girls. Although some of the studies just discussed reported in-group math preferences among young girls (ages 6–11) [28–30,32,33], others found that young girls hold egalitarian views [27] or even endorse stereotypes that boys are more able than girls in math [31]. The findings also do not uniformly support other explanations such as social status, cultural stereotypes, or actual differences in performance as factors influencing children's beliefs. Findings with older children are even less straightforward. Some studies show older boys and girls (ages ranging from 12–15) favor their own gender, some find youth are egalitarian, and yet others find that they favor the outgroup, with no clear age trends [30–32,34–37].

Research results regarding boys' and girls' gendered beliefs about language ability are also mixed. To our knowledge, no studies have reported girls favoring boys in language domains across any age range, but a few studies have found that young girls (ages ranging from 6–11) reported no differences between boys and girls in their liking of, or competence in, language [27,38]. Italian first grade girls did not show a gender preference when choosing between a picture of a boy and that of a girl when selecting who was better at language [38]. Singaporean girls in first, third, and fifth grade also did not show a gender preference when rating boys' and girls' liking of language [27]. Boys, on the other hand, more consistently endorse the female-language stereotype, but exceptions have been found whereby boys report egalitarian beliefs or even favor boys in language. For instance, first grade Italian boys did not show a gender preference when choosing between a picture of a boy and one of a girl when selecting who was better at language [29,38]. In a U.S. sample, fourth grade boys were egalitarian regarding their beliefs about who was better at reading and writing [32]. Therefore, the influences of in-group preferences versus cultural stereotypes, social status, or other factors on children's explicit beliefs about math and language abilities remain unclear.

A problematic aspect of research on academic stereotypes focused on explicit beliefs is that it relies on self-report measures, which allow participants to control their responses. Children may report beliefs that match perceived social expectations or self-presentation goals, regardless of their own beliefs. For instance, 6- to 8-year-old children who thought they were being videotaped suppressed their explicit in-group preference and outgroup prejudice, whereas those who thought the camera was off did not [39]. The mixed findings in explicit attitudes,

then, might reflect variability in children's personal beliefs, but they could also be an artifact of children's reactance to being questioned about a socially sensitive topic.

Examining children's implicit biases circumvents the self-presentation problems associated with explicit measures. Implicit biases are automatic associations that are measured using cognitive tests that capitalize on reaction times or priming procedures. Responses to these tests are difficult to control and thus, are often independent of intent [40–42]. Such associations are not necessarily indicative of explicit beliefs, and may even be inconsistent with them [43]. Although explicit measures are informative in their own right, implicit measures can provide additional information regarding the impact of socially sensitive topics, such as stereotypes, that may not be readily accessible through explicit reports. In this study, we explored both children's explicit beliefs and their implicit biases regarding math and language ability among boys and girls. Our hypotheses regarding explicit beliefs were that in-group preferences would be evident in the youngest age group (elementary school children) regarding math ability. We did not have specific hypotheses regarding older children, as multiple factors can affect their responses on explicit tests, including societal stereotypes, social status differences, and social desirability, among others. Regarding language ability, we hypothesized that both boys and girls would favor girls because those beliefs are congruent with multiple factors, including societal stereotypes, actual performance, and social acceptability of endorsing the stereotype (i.e., the stereotype is less condemned).

## Children's implicit biases about academic abilities

The majority of studies examining children's gender implicit biases regarding academic abilities have measured implicit bias using the Implicit Association Test (IAT). This test measures reaction times in stereotype-congruent versus stereotype-incongruent conditions, the theoretical principle being that children (and adults) respond more quickly to paired categories that are cognitively associated. More specifically, in the stereotype-congruent condition, children press one computer key if they see words associated with boy names or math concepts and another key if they see words associated with girl names or language concepts. Faster reaction times within this stereotype-congruent condition—compared to the stereotype-incongruent condition in which paired categories are boys/language and girls/math—are typically interpreted as indicating math-male implicit bias. Using this procedure, several studies have found that girls show stronger stereotypic implicit biases than boys. In a sample of Italian children in first grade, boys did not show an implicit bias whereas girls showed stereotype-consistent implicit bias [28]. Using a paper-and-pencil IAT in which participants had 30 seconds to classify words into categories, Italian girls in Grades 3, 5, and 8 showed an implicit bias, such that they categorized more words in the stereotype-congruent condition compared to the stereotype-incongruent condition [30]. Boys, on the other hand, showed stereotypic associations in eighth grade, but not in third or fifth grade. In a sample of German children in Grades 4, 7 and 9, boys did not show any stereotypic associations across any age group, whereas girls in Grades 4 and 9 showed significant stereotyping [31]. In a rare exception for this literature, girls in seventh grade did not show any bias. These are just a few examples, but other studies show similar patterns: Girls demonstrate a stereotypic implicit bias, whereas boys either demonstrate a stereotypic implicit bias or show no bias at all [27,29,35,38,44,45]. These findings cover age spans from 5- to 15-years-old, and use samples across multiple nations, including Canada, Chile, Italy, Singapore, and the United States. Based on the results of these IAT studies alone, one conclusion is that children (girls in particular) assimilate societal stereotypes about gender differences in math ability favoring boys from an early age [45]. This interpretation implies that research efforts and interventions ought to be focused on children's *math* associations and

beliefs. Yet, an outstanding question is whether these biases are truly about math rather than language, even though they are typically labeled as math-male biases.

Due to the paired nature of IAT categories, it is impossible to disambiguate whether children who show a "male-math" bias do so because they associate math with boys more strongly than they associate math with girls, or because they associate language with girls more strongly than they associate language with boys, or both. In some cases, studies have found that IAT scores predict math outcomes, but even then what is labeled as a "math outcome" is a relative difference between math and language outcomes. For instance, in a German sample of 4th, 7th, and 9th graders, stereotype-consistent IAT scores were correlated positively for girls and negatively for boys with intentions to enroll in language versus math courses, grades in language versus math, and self-concepts in language versus math [31]. One experimental study did find that IAT scores were positively correlated with math performance on a test, explaining part of the relation between exposure to the math-male stereotype and math performance [28]. However, the relation to math outcomes is not always consistent, as some studies have found no relation [30,35].

A recent study further challenges the assumption that math associations drive previous findings. Critically, the study used an implicit bias measure that did not confound math and language bias [36]. Instead, participants pressed one key if they saw positive adjectives related to "doing good work in mathematics" and another key if they saw negative adjectives related to "doing poor work in mathematics." In the stereotype-congruent condition, the key corresponding to doing good work in math was on the same side as a smiley face and a picture of a male doll on the screen. The key for doing poorly in math was paired with a sad face and a picture of a female doll. The pairings were reversed in the stereotype-incongruent condition. In a different set of blocks altogether, language implicit biases were assessed using an identical procedure, but the adjectives were described as being related to doing good or poor work in reading. Implicit biases were scored as the difference in reaction times between stereotype-congruent and stereotype-incongruent trials. Thus, the researchers were able to obtain implicit bias scores for math separately from those for language. Using this procedure in a sample of Canadian children in Grades 4–6, researchers found—in stark contrast to previous results using the IAT—that girls held a counter-stereotypical implicit bias favoring girls over boys in math. Boys demonstrated no math implicit bias. Language biases, on the other hand, were consistent with previous IAT findings; girls showed a stereotypical language-female bias, whereas boys demonstrated no language implicit bias.

In the only other study (to our knowledge) that used an implicit bias measure that disambiguated math and language implicit biases, girls also did not demonstrate stereotypical math-male bias [46]. In that study, German students in Grade 9 completed a go/no go association task as the measure of implicit bias. This task required participants in the stereotype-congruent condition to press the space bar if they saw words on the computer screen associated with math or boys and to ignore other words (i.e., words associated with girls and neutral stimuli). In the stereotype-incongruent condition they responded to words associated with math or girls and ignored all other words. The same procedure was applied to measure language bias. Scores were based on reaction time differences between the stereotype-congruent and stereotype-incongruent blocks. The results were that girls did not show a math-male bias, whereas boys did. On the other hand, girls showed an implicit language-female bias, whereas boys showed a counter-stereotypical bias favoring boys in language.

To summarize, all studies examining math/language implicit biases using the IAT have found that girls demonstrate stereotypical implicit biases (with the exception of one seventh grade subsample). Although these results might reflect math-male biases, language-female biases, or both biases, they have been labeled as "math-male" biases. In contrast, the only two

published studies that have employed implicit bias measures that disambiguate math and language biases have not supported the notion that girls hold math-male biases. We base these statements on a literature search for peer-reviewed articles on Google Scholar and PsychINFO using different combinations of keywords: "IAT," "implicit bias," "implicit stereotype," "math," "language," "children," and "girls" (up-to-date as of March 2020). This pattern of findings suggests that measurement differences could explain the seemingly conflicting results. Further, when understood from the perspective of language gender biases, all the findings align.

In our study, we also used a measure that disambiguates math and language implicit biases. We tested two different predictions based on two theoretical accounts. The first was that girls would hold math-male implicit biases, consistent with the idea that they assimilate cultural stereotypes favoring boys in math (in line with the interpretation of IAT results). The alternatively hypothesis was that girls would show a math-female counter-stereotypical bias, consistent with national gender differences in overall academic performance. Because performance and stereotypes about language are consistent with each other, we expected girls to show implicit biases favoring girls in language. We were agnostic about the implicit biases of boys, as previous findings have been inconsistent and do not clearly favor one theoretical account over another. The dissociative processes by which girls and boys form automatic associations is in itself interesting, but not the subject of this report.

## Affect misattribution procedure

In the current study, we used the Affect Misattribution Procedure (AMP) [47] to measure implicit biases about math and language ability among girls and boys. The AMP has several strengths for studying implicit biases [48]. First, it tests automatic associations to single domains, meaning it can assess implicit biases regarding math and language separately. Second, it has a simple structure, which is ideal for use with children. Participants follow simple instructions to make binary judgements about ambiguous stimuli across several trials. In the present study, the binary judgements were "good at math" versus "bad at math" (or language arts), and the stimuli were Chinese symbols. Each ambiguous stimulus (i.e., Chinese symbol) is preceded by a prime that participants are instructed to ignore. In the present study, the prime was a picture of a boy or a girl. The AMP measures participants' unintended misattribution of affect or semantic content (e.g, [49,50]) from the prime to the ambiguous stimulus. For example, when asked to judge whether a Chinese symbol means "good at math" or "bad at math," participants unintentionally use their judgements about the preceding prime (e.g., photo of a girl/boy) to make a response.

One potential concern about the AMP is that participants could ignore instructions and directly rate the primes, making the measure more akin to an explicit measure. To address this concern, one study tested two different AMP conditions assessing race implicit bias, one where adult participants were instructed to directly rate the primes and another one where they were instructed to ignore the primes and rate the target stimuli [51]. These two conditions yielded divergent results. The traditional AMP predicted racial bias in an impression formation task, whereas the "explicit" AMP did not, presumably because individuals were motivated to control expressions of prejudice in the explicit condition. Indeed, motivation to control prejudice was related to the explicit AMP but not the traditional one. These results suggest that, under normal conditions (i.e., when people are instructed to ignore the photo primes and evaluate the symbols), participants are not intentionally rating the photo primes. Though these findings were based on adult samples, they likely extend to children, as children tend to have less self-regulatory skills than adults to control their responses [52]. Another study addressing

this concern found that adults are not able to accurately introspect on what influenced their response pattern [51], rendering self-reports of this information unreliable. Taken together, these findings suggest that any systematic influence of the photo primes on the ratings of the ambiguous stimuli reflects an unintended/automatic, and thus implicit, bias.

The potential weaknesses of the AMP have been tempered by empirical evidence, and its strengths have been documented extensively. Another strength of the AMP is its reliability. Meta-analytic procedures have shown the AMP to be more reliable than most reaction-time-based measures [48]. There is also evidence that it is a reliable measure for use with children, with good predictive validity [53].

Use of the AMP in this study allowed us to examine gender and age differences in implicit gender biases regarding math and language ability separately. Given the potential methodological specificity of previous findings regarding math implicit biases in children, it was important to select a measure that not only disambiguates biases by domain, but is also simple enough to use with children, reliable, and has proven to be valid.

## Relations between implicit biases and explicit beliefs

In addition to measuring age and gender differences in reports, we also measured relations between implicit biases and explicit beliefs. Explicit reports are presumed to reflect personally endorsed attitudes, which may be shaped by motivated reasoning such as social desirability or the need to protect one's social identity. Recent theoretical perspectives on implicit biases, in contrast, suggest that situational effects are a strong determinant of implicit biases [54]. Examples of these situational effects are cultural stereotypes cued by media or perceptible environmental inequalities, such as differences in classroom performance. Thus, for example, an adolescent who is aware of the math-male stereotype might show implicit biases favoring boys in math even if she does not personally endorse the stereotype, simply because it has been activated by something in the environment. Similarly, a child who observes differences in classroom performance might automatically associate girls with academic success, but still assert that boys are better than girls at a given subject in order to protect his gender self-esteem.

Thus, implicit and explicit measures might yield unrelated results because of motivational biases operating in explicit reports, or because implicit bias responses reflect activation of cultural knowledge or environmental cues not endorsed or acknowledged by the individual. Meta-analyses examining relations between the two have shown small, positive relations, with mean effect sizes often ranging between .20 and .24 [43,55]. Results of individual studies vary widely, however, with the strength of relations shaped by moderators such as conceptual correspondence between the two measures and other task characteristics. Findings from investigations focusing specifically on children's academic biases and beliefs have also found small to no correlations. In their study of Singaporean children in first, third, and fifth grades, Cvencek et al. [27] found very low to no correlations between children's implicit and explicit reports of gender differences in math ability. Implicit attitudes were unrelated to explicit math stereotypes in Passolunghi et al.'s [30] study of third, fifth, and eighth grade Italian children. Though these results could be indicative of a dissociation between implicit biases and explicit beliefs among children, they could also be indicative of the lack of correspondence between the IAT, which confounds math and language and was the implicit measure used in those studies, and their explicit measures, which focused on math.

## Current study

The aim of this paper was to examine age and gender differences in implicit biases and explicit beliefs regarding gender differences in math and language abilities. By using the Affect

Misattribution Procedure (AMP) [47,48] to measure implicit biases, we were able to obtain independent measures of gender biases regarding math as opposed to language abilities. We used a cross-sectional sample of youth in elementary school, middle school, and high school to test age differences.

With regard to explicit stereotypes, we expected age differences with the youngest age group most likely to favor their own gender in math. We did not have specific predictions about older children, as we could envision multiple factors influencing their beliefs, including the substantial efforts in recent decades to encourage the idea that girls can excel in math, increasing sensitivity to such social norms with age, cultural stereotypes, social status differences, and actual differences in performance. In contrast, given cultural stereotypes emphasizing girls' success in language domains, combined with gender differences in academic performance, we expected that youth of both genders would favor girls in their explicit reports of language abilities.

With regard to youth's implicit biases, we envisioned two potential outcomes based on two different theoretical accounts. We expected girls would show traditional math-male biases if they have assimilated cultural stereotypes that favor boys in math. In contrast, girls would favor girls in math if pervasive differences in academic performance are the primary factor shaping automatic associations about gender and math ability. We expected girls to show implicit biases favoring girls in language, regardless, as both cultural stereotypes and differences in academic performance favor girls in language domains. We were agnostic about trends for boys.

With regard to relations between implicit and explicit measures, given developmental differences across this age range and results of prior studies, we expected weak positive relations between explicit and implicit measures in each domain, with the possibility that the strength of relation might decrease with age. Whereas younger youth may be more transparent and explicitly report their automatic associations, older youth might control their responses so that their implicit biases are more dissociated from their explicit reports.

## Method

### Participants

Participants were 270 youth (141 girls) ages 8 to 15. Youth were recruited from public libraries and schools in the southeastern region of the United States. A sensitivity analysis conducted in G*Power (Version 3.1.9.2) [56] indicates that this sample is sufficient to detect main effects as small as $\eta^2 = .03$ (f = .17) and interaction effects as small as $\eta^2 = .04$ (f = .19) at .80 power.

Youth ages 8–10 were grouped into an elementary school category ($n$ = 101, 53 girls and 48 boys, $M_{age}$ = 8.9, $SD$ = 0.7). Those ages 11–13 were grouped into a middle school category ($n$ = 67, 25 girls and 42 boys, $M_{age}$ = 11.5, $SD$ = 0.7), and youth ages 14–15 were grouped into a high school category ($n$ = 99, 63 girls and 36 boys, $M_{age}$ = 14.4, $SD$ = 0.8). Our sample was 49.1% White, 29.9% Black, 12.0% Hispanic, 5.2% mixed race/ethnicity, 2.2% Asian, and 0.37% other (1.23% did not report their race).

### Procedures

All procedures were consistent with ethical standards of the American Psychological Association and were approved by the Institutional Review Board at the University of North Carolina at Chapel Hill. After parents provided informed consent, youth gave verbal and written assent to participate. Next, youth completed the Affect Misattribution Procedure (AMP), an implicit measure of academic stereotypes. A researcher read the initial instructions aloud and gave the participant intermittent reminders. The AMP is a computerized task that was administered on a laptop computer. Finally, participants completed a paper survey that included an explicit

measure of academic stereotypes, along with other measures not included in the current report. If needed, a researcher assisted younger participants in reading the instructions and questions, but to prevent social desirability effects, the researcher read from a different survey, facing away from the child in order not to look at his or her responses. The research team included both men and women, African Americans, non-Hispanic Whites, and Hispanics. All stimulus materials reported here can be found in S1 File.

Children and adolescents were recruited from schools and from the community through announcements posted in public locations. Data collection was conducted in public libraries, at a local YMCA, and in four participating schools. In each of those settings, testing took place individually in a quiet and secluded area, facing away from other people.

## Measures

**Implicit bias.**   We used the Affect Misattribution Procedure (AMP) [48] to measure implicit bias. The AMP has been validated as a measure of implicit biases in adults [47] and also in children ranging from 4 to 12 years old [57–59]. Each trial of the AMP began with a brief presentation (200 ms) of a photo on the computer screen. The photographs used as primes included 40 images of early adolescents—20 girls and 20 boys, balanced in race (Black and White). The photos were selected based on a pilot study to ensure that photos of the two genders did not differ on perceived attractiveness, age, or mood. Internal consistency for the measure was $\alpha$ = .40 (procedure outlined in [47], Experiment 1).

Following the randomly selected photograph, a black and white pattern (125 ms) and a Chinese symbol were presented (150 ms). A black and white pattern then appeared until the participant made a response. Participants were instructed to ignore the photograph and make a judgment about the meaning of the Chinese symbol. They made these judgements in two blocks of trials, one for math and one for language. Each block consisted of 40 trials. For example, in one block participants guessed whether each symbol was a word related to the ideas of "good at math" versus "bad at math." In the next block, they guessed whether each symbol was related to being "good at language arts" versus "bad at language arts." The keys on the keyboard were clearly labeled "good" or "bad" (in place of the "F" and "J" keys). Participants were told that each symbol was "a word from the Chinese alphabet," and that we were interested in how people make guesses about the meaning of words. The instructions further specified that the participant should rate about half of the symbols as good at math (or language arts) and half as bad at math (or language arts). Two other school domains (sports and science) were included in separate blocks, but are not the focus of this report. Block presentation was counterbalanced across participants. We do not have information about whether any participants were familiar with the Chinese symbols in our study or on their thoughts about the cover story. However, we should note that the believability of the cover story is superfluous to the task's objective, which was to rate ambiguous stimuli preceded by a prime. The mechanism by which the AMP functions (i.e., misattribution of affect/semantic content from prime to target) does not depend on the construal of the task. Familiarity with the Chinese symbols does present a problem, which is that the symbols would no longer be neutral, weakening priming effects. Thus, though unlikely given our sample's demographics, we should note that the effects reported here might be underestimates.

Participants were reminded between blocks that they should ignore the photo of the person and to just focus on the symbol. The sequence of photographs was randomized within each block, and the sequence of domains (language, math) was randomized across participants.

Each participant had two implicit bias scores for each domain, one representing their implicit bias regarding girls (i.e., the proportion of times the student designated "good in

math" [or language] after seeing a photo of a girl) and one representing their implicit bias regarding boys (i.e., the proportion of times the student selected "good in math" [or language] after seeing a photo of a boy). Because these scores were proportions of the total number of times they viewed stimuli of each gender, scores had the possible range of zero to 1.00, with higher scores indicating greater bias favoring the gender of the prime photograph, which we refer to as the prime gender.

**Explicit beliefs.**   Youth used a visual analog scale (VAS), consisting of a 100 mm horizontal line, to indicate with a vertical mark how well they thought boys or girls performed on a specific academic subject and how difficult they thought boys or girls found the subject. They could place a mark anywhere on the line, which allowed them to give very low or high ratings without having to choose the extreme option, as is the case with Likert scales. This attribute of VAS lines is important when measuring beliefs or attitudes that are sensitive to social desirability effects, such as stereotypes.

Participants answered two items regarding math and two items regarding language ability (e.g. *I think that in MATH boys do this well. . .*, and *I think that boys find MATH. . .*, with the extremes of the line labeled from "not well at all" to "very well" and from "very hard" to "very easy," respectively). Each item was answered separately in regard to boys and girls, and each gender group was represented on a separate page. Youth rated the competence of boys and girls in other domains, both academic and non-academic, but those data are not the focus of this report. Items were scored by measuring the distance in millimeters from the left scale anchor to the line drawn by the respondent for each item. The two items corresponding to each subject and gender were averaged. Scores ranged from zero to 100, and the correlations between the two items in each measure (i.e., math and language) ranged from $r = .53$ to $r = .61$. Higher values indicate endorsement of greater competence in math/language.

## Results

All data for this study can be found in the online repository: https://osf.io/fv5h8/. Our exclusion criteria for the implicit bias task (pressing the same key on all trials or alternating keys on all trials) did not apply to any participant.

### Implicit gender bias

Tables 1 and 2 show the average proportion of prime photos that youth associated with "good at math" or "good at language," respectively, split by prime gender and participant characteristics. To assess implicit biases regarding math and language ability in boys and girls, we conducted a 2(Participant Gender) x 3(Age Group) x 2(Academic Subject) x 2(Prime Gender) ANOVA, with Participant Gender and Age Group as between-subject factors, Academic Subject and Prime Gender as within-subjects factors, and implicit scores as the dependent variable.

The main effect of Prime Gender was significant, $F(1, 256) = 6.79$, $p = .010$, $\eta^2 = .03$, and was qualified by a significant Participant Gender x Prime Gender interaction, $F(1, 256) = 10.94$, $p = .001$, $\eta^2 = .04$. Fig 1 displays that girls of all ages showed an implicit own-gender bias. Girls rated a greater proportion of symbols preceded by photos of girls as good at math and language arts compared to symbols preceded by photos of boys, and this difference was statistically significant (see Table 3). Boys, on the other hand, did not show implicit bias. The Participant Gender x Academic Subject x Prime Gender interaction was not significant, $F(1, 256) = 0.62$, $p = .432$, failing to provide evidence that boys and girls differed in the extent to which they showed consistent gender-domain associations. None of the other interactions were significant. These results show implicit biases as being invariant across age, with girls

**Table 1. Math implicit bias scores by age group, participant gender and prime gender.**

| Age Group | Participant Gender | Prime gender | | | | |
|---|---|---|---|---|---|---|
| | | Girls | | Boys | | |
| | | *Mean Proportion "Good at math"* | *SD* | *Mean Proportion "Good at math"* | *SD* | *n* |
| Elementary School | Girls | .541 | .122 | .478 | .126 | 51 |
| | Boys | .527 | .124 | .547 | .097 | 47 |
| Middle School | Girls | .571 | .107 | .479 | .132 | 24 |
| | Boys | .524 | .086 | .525 | .114 | 42 |
| High School | Girls | .558 | .129 | .527 | .130 | 62 |
| | Boys | .543 | .108 | .539 | .130 | 36 |

*SD* = standard deviation. Girls at all three ages showed implicit bias favoring girls in math.

holding an implicit in-group bias for both math and language, and boys not associating either math or language with either gender preferentially.

An alternative interpretation of our results is that the AMP was measuring generalized gender biases (e.g., girls = good) rather than domain-specific biases (e.g., girls = good at math). To test this possibility, we examined girls' and boys' implicit biases regarding sports ability (one of the domains included in the study, but not the focus of this report). We report the results in our S1 File. In short, boys showed evidence of implicit bias favoring boys for the sports domain, but girls showed no implicit bias. Although these results might still reflect generalized biases (i.e., girls = good at academics / boys = good at athletics), they suggest that the AMP was sensitive to domain category and not just valance. In theory, if the instrument distinguishes between broad categories (e.g., sports versus academics), then it can distinguish between the categories specified in the actual task (sports v. math v. language), unless the effects of the specific academic subjects are so small as to be overwhelmed by the broader category. This latter point is in itself informative and contrary to the current understanding of girls' implicit biases.

Overall, these results challenge the assumption that girls automatically associate math ability with boys rather than girls, and they suggest that measurement differences could explain ostensibly conflicting results from previous studies. Of course, more research is needed in this area to be conclusive, but the current study highlights the importance of measuring math and language implicit biases separately to better understand children's automatic associations between gender and domain-specific abilities. Especially in light of persistent gender disparities in STEM, this issue deserves careful attention, as children's perceptions of language domains may have implications for the interests they develop and the trajectories they pursue.

**Table 2. Language implicit bias scores by age group, participant gender and prime gender.**

| Age Group | Participant Gender | Prime gender | | | | |
|---|---|---|---|---|---|---|
| | | Girls | | Boys | | |
| | | *Mean Proportion "Good at language arts"* | SD | *Mean Proportion "Good at language arts"* | SD | *n* |
| Elementary School | Girls | .553 | .121 | .473 | .124 | 51 |
| | Boys | .532 | .124 | .518 | .132 | 47 |
| Middle School | Girls | .544 | .132 | .548 | .136 | 24 |
| | Boys | .500 | .109 | .520 | .086 | 42 |
| High School | Girls | .569 | .126 | .527 | .123 | 62 |
| | Boys | .560 | .125 | .572 | .144 | 36 |

*SD* = standard deviation.

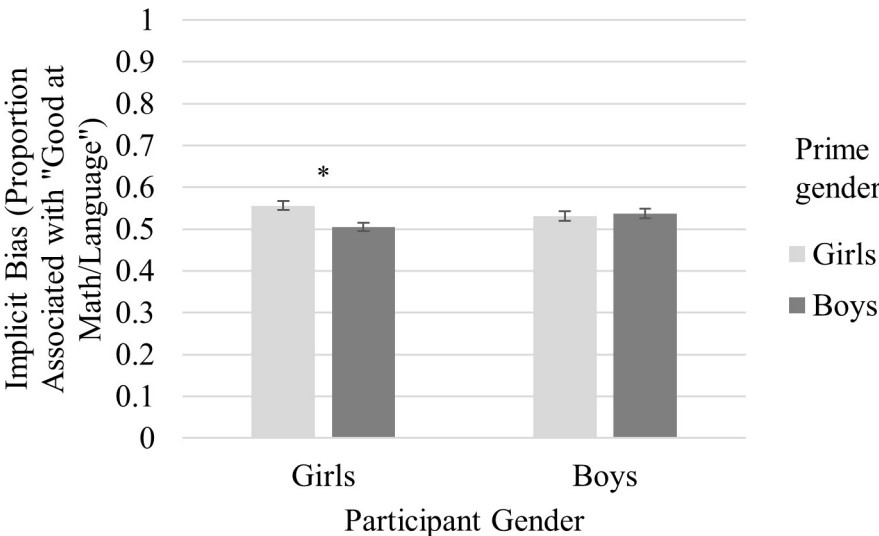

**Fig 1. Estimated marginal means for implicit bias scores by participant gender.** Values indicate the proportion each prime gender associated with "good at math" or "good at language arts." Bars represent standard errors.

## Explicit gender beliefs

Tables 4 and 5 show the average competence scores that children gave to boys and girls, split by participant characteristics. To assess gender differences in children's explicit beliefs regarding math and language ability in boys and girls, we conducted a 2(Participant Gender) x 3(Age Group) x 2(Academic Subject) x 2(Target Gender) ANOVA, with Participant Gender and Age Group as between-subject factors, Academic Subject and Target Gender as within-subject factors, and explicit scores as the dependent variable. We found significant main effects of Academic Subject, $F(1, 258) = 11.21$, $p = .001$, $\eta^2 = .05$, Target Gender, $F(1, 258) = 57.64$, $p < .001$, $\eta^2 = .18$, and Age Group, $F(2, 258) = 6.71$, $p = .001$, $\eta^2 = .05$. These main effects were qualified by significant two-way and three-way interactions. We only describe the three-way interactions here, as they qualify the two-way interactions (see the S1 File for full results). The Age Group x Participant Gender x Target Gender interaction was significant, $F(2, 258) = 6.82$, $p = .001$, $\eta^2 = .05$. Adjusting for multiple comparisons using a Bonferroni adjustment, we found that with scores collapsed across the two academic domains, girls of all three age groups favored girls, whereas boys favored girls in middle school and high school, but not in elementary school (statistics appear in Table 6).

The Age Group x Academic Subject x Target Gender interaction was also significant, $F(2, 258) = 3.51$, $p = .031$, $\eta^2 = .03$. Children favored girls in language across all three age groups. In contrast, in the case of math, only children in elementary school showed a bias, favoring girls

**Table 3. Pairwise comparisons of math implicit bias scores by participant gender.**

| Participant Gender | Mean Diff Prime Gender (Girls—Boys) | SE | p | 95% CI | n |
|---|---|---|---|---|---|
| Girls | .051* | .012 | .000 | [.026, .075] | 137 |
| Boys | -.006 | .012 | .614 | [-.030, .017] | 125 |

*Diff* = difference, *SE* = standard error, *p* = probability value, *CI* = confidence intervals, *n* = number of participants. Adjustment for multiple comparisons: Bonferroni. Mean differences represent the proportion of symbols rated "good at math/language" when the prime was a girl minus the proportion when it was a boy.

**Table 4. Math explicit belief scores by participant gender, age group, and target gender.**

| Age Group | Participant Gender | Target gender | | | Target gender | | |
|---|---|---|---|---|---|---|---|
| | | Girls | | | Boys | | |
| | | *Mean math competence* | *SD* | | *Mean math competence* | *SD* | *n* |
| Elementary School | Girls | 74.94 | 18.53 | | 57.43 | 26.83 | 53 |
| | Boys | 72.32 | 16.88 | | 73.10 | 18.17 | 47 |
| Middle School | Girls | 63.05 | 17.59 | | 61.66 | 16.93 | 25 |
| | Boys | 72.48 | 17.70 | | 65.21 | 14.32 | 42 |
| High School | Girls | 67.23 | 17.90 | | 63.65 | 16.65 | 62 |
| | Boys | 61.87 | 9.80 | | 59.95 | 10.42 | 36 |

*SD* = standard deviation, *n* = number of participants.

over boys in math. Youth in middle school and high school did not show a gender bias in explicit reports of math ability. These results should be interpreted with caution, though, as the effect size is below the threshold calculated by our sensitivity analysis.

Though the four-way interaction was not significant, perhaps due to low power, we conducted pairwise comparisons to directly test our hypothesis that the youngest children would favor their own group in math. We report our results in the S1 File. Our hypothesis was partly supported; elementary-school girls reported girls as being better than boys at math. Elementary-school boys were neutral regarding gender differences, but they gave boys a significantly higher rating in math than elementary-school girls gave boys. Our hypothesis that children of all age groups would favor girls over boys in language ability was also partly supported; with the exception of elementary-school-aged boys (who were neutral), children favored girls over boys in language ability.

In summary, regardless of age group, girls explicitly endorsed an in-group preference in both math and language. Boys, on the other hand, explicitly favored girls over boys in math and language only later in development (i.e., middle school and high school, but not elementary school). These results are somewhat consistent with what we predicted. We hypothesized that younger children would show an in-group preference in math. Though that was not exactly the case for boys, younger boys were, on average, less prone than older boys to explicitly favor girls. Unexpectedly, older children of both genders rated girls' ability in math and language as superior to that of boys. Their agreement on these gender differences could be due to their observations of academic performance within their classrooms and schools.

**Table 5. Language explicit belief scores by age group, participant gender and target gender.**

| Age Group | Participant Gender | Target gender | | | Target gender | | |
|---|---|---|---|---|---|---|---|
| | | Girls | | | Boys | | |
| | | *Mean language competence* | *SD* | | *Mean language competence* | *SD* | *n* |
| Elementary School | Girls | 77.30 | 20.06 | | 58.96 | 27.11 | 53 |
| | Boys | 76.43 | 16.24 | | 73.40 | 18.92 | 47 |
| Middle School | Girls | 76.03 | 15.13 | | 61.66 | 18.59 | 25 |
| | Boys | 75.29 | 18.39 | | 65.67 | 15.80 | 42 |
| High School | Girls | 72.59 | 14.87 | | 58.97 | 15.15 | 61 |
| | Boys | 70.00 | 12.50 | | 54.06 | 13.96 | 36 |

*SD* = standard deviation, *n* = number of participants.

**Table 6. Pairwise comparisons of explicit scores by age group and participant gender.**

| Participant Gender | Age group | Mean Diff (Girls—Boys) | SE | p | 95% CI | n |
|---|---|---|---|---|---|---|
| Girls | Elementary School | 17.92* | 2.49 | < .001 | [13.02, 23.43] | 53 |
| | Middle School | 7.88* | 3.62 | .030 | [-0.75, 15.01] | 25 |
| | High School | 8.61* | 2.32 | < .001 | [4.04, 13.17] | 63 |
| Boys | Elementary School | 1.13 | 2.64 | .670 | [-4.07, 6.33] | 48 |
| | Middle School | 8.45* | 2.79 | .003 | [2.94, 13.95] | 42 |
| | High School | 8.93* | 3.02 | .003 | [2.98, 14.87] | 36 |

*Diff = difference, SE = standard error, p = probability value, CI = confidence intervals, n = number of participants. Adjustment for multiple comparisons: Bonferroni. Mean differences represent how much more competent participants rated girls to be in math and language compared to boys. Asterisks indicate that girls were rated as more capable than boys.*

### Correlations between explicit and implicit measures

Finally, we computed bivariate correlations between implicit bias and explicit stereotypes. Implicit math gender biases were not correlated with explicit math stereotypes, $r(259) = -.03$, 95% $CI = [-.15, .09]$, $p = .624$. Neither were implicit language biases correlated with explicit language stereotypes, $r(258) = .11$, 95% $CI = [-.01, .23]$, $p = .076$. Bivariate correlations of implicit and explicit stereotype scores for each domain, calculated separately for each participant gender and age group revealed a similar pattern after adjusting the alpha level to .008 for multiple tests. Implicit biases and explicit stereotypes were not correlated for math or language across any age group for either boys or girls (all $p$'s > .008).

Correlations between pairs of implicit scores and pairs of gender group competence scores are presented in Table 7 (correlations split by participant gender are reported in the S1 Table in S1 File, but in general, they follow the same pattern). For these correlations, for implicit scores we used the proportion of items in which [girls; boys] were associated with the "good in" prompt; for explicit group competence, we used the average of the two explicit items for each gender. All explicit gender group competence ratings were positively associated. For example, youth who rated boys as highly competent in math also tended to rate boys as highly competent in language, and youth who rated boys as competent in language tended to also rate girls as capable in language, $r$'s = .672 and .328, respectively. In contrast, implicit scores were positively correlated only within gender.

### Discussion

Because of the salience of gender identity for most children and adolescents, youths' perceptions of gender differences in academic skills are posited to shape perceptions of the self, classroom motivation and behaviors, and long-term career goals [1,2,60]. In the current study, we examined explicit beliefs and implicit biases regarding perceptions of gender differences in

**Table 7. Correlations among implicit bias scores (above the diagonal) and explicit gender group competence (below the diagonal).**

| | Math-Boys | Math-Girls | Language-Boys | Language-Girls |
|---|---|---|---|---|
| Math-Boys | — | .021 | .281*** | .111 |
| Math-Girls | .148* | — | .016 | .277*** |
| Language-Boys | .672*** | .328*** | — | -.027 |
| Language-Girls | .308*** | .499*** | .332*** | — |

N = 263 for implicit bias scores and N = 267 for explicit stereotypes.

math and language abilities in a cross-sectional sample of youth from elementary school, middle school, and high school. In addition to testing possible age and gender differences, we also measured correlations between implicit and explicit gender stereotypes. We found that girls showed in-group preferences in math and language across both implicit and explicit measures. These findings are consistent with gender differences in academic performance at the national level, and they contradict traditional math stereotypes. Boys, on the other hand, showed no implicit biases within any of the three age groups. Explicitly, boys in middle school and high school favored girls in math and language. Boys in elementary school reported egalitarian beliefs on the explicit measure.

The results of this study, at first glance, appear to be at odds with persistent gender disparities in STEM careers. They also contradict previous implicit bias findings that have shown math implicit biases among girls. In the next sections, we discuss potential explanations for these results and their implications.

## Children and adolescents' implicit biases regarding gender differences in abilities

An important contribution of the current study is that we used a measure of implicit gender biases in which math and language abilities were not confounded. Most prior research in this area has used the Implicit Association Task, in which the two pairings (e.g., boys-math; girls-language) are not measured independently. Of note, authors of those studies have often interpreted their results as indicating an implicit bias favoring boys in math, and have used their measures to predict math-related outcomes. However, our results suggest that girls hold implicit biases favoring girls in both math and language. In supplemental analyses, we examined whether these associations might have been an artifact of the measure's sensitivity, which may not have been granular enough to test domain-specific associations, but only broader associations (e.g., "girls = good" rather than "girls = good at math). We examined implicit biases regarding sports ability and found a different pattern of results from academic biases. Girls were neutral whereas boys favored boys in sports ability. Although these biases may still reflect a general association between girls and good academic performance, they suggest the AMP was sensitive to domain-category and challenge prior assumptions that girls implicitly favor boys over girls in math. At a minimum, our study suggests that gender math associations do not override the positive associations that girls have about girls' general academic performance compared to boys.' Another interpretation is that girls' associate girls more than boys with good math performance, in line with their explicit reports.

It is possible that a strong girls-language association—an association that may be more likely to emerge on both implicit and explicit measures (rather than just implicit) due to its relative social acceptability—may be a particularly powerful predictor of both better language outcomes and, perhaps, worse math outcomes. To our knowledge, our study is only the third to use an implicit bias measure that disambiguates math and language biases allowing us to better assess these nuanced research questions. In the two earlier studies, one study found a counter-stereotypical math-female bias among girls [36], and the other found no math bias among girls [46]. Measurement differences could account for these ostensibly conflicting findings in the literature, and future research should continue to disentangle gendered associations within different domains to clarify their relation to meaningful academic outcomes.

Using the AMP, we found that in contrast to explicit reports, which differed across age, gender, and academic domains, youths' implicit biases differed primarily by gender of the respondent, with girls favoring girls in both domains and boys showing egalitarian responses. These responses to our implicit bias measure might reflect a combination of in-group preference as

well as youths' lived reality of gender differences in school performance. Beginning with school entry and continuing throughout primary and secondary education, girls receive better grades than boys, are rated by teachers and parents as more engaged in schoolwork, and have higher graduation rates [21,24,25]. In addition, some scholars have suggested there is a discord between traditional norms of masculinity and behaviors that promote academic success such as help-seeking and cooperation [61,62]. These factors have led scholars to posit that school is perceived as a feminine domain. Indeed, using an implicit measure, Heyder and Kessels [63] found that German ninth graders associated school with girls more strongly than with boys, and that boys' tendencies to view school as feminine and to ascribe negative masculine traits to themselves were related to lower grades in German. The view of academic success as a feminine trait may have led girls in the present study to show implicit biases favoring girls in both domains, whereas for boys, those views may have been tempered by a tendency to show in-group preference, resulting in their egalitarian scores on the task.

Though, at first glance, our results might not appear consistent with gender disparities in STEM careers, they are revealing in that they support recent theoretical frameworks suggesting that girls opt out of math, not due to perceived deficit in math ability compared to boys, but due to perceived strength in language ability over math ability. For example, a large international study of 15-year-old students found that girls' comparative advantage in reading as opposed to math can largely explain gender disparities in intentions to pursue math-related careers [64]. In that study, girls who were found to be good at math were more likely than boys to be even better at reading than at math. The gap between math and reading performance accounted completely for gender differences in math self-concept, interest in math, and attitudes towards math. Other studies have also found that intra-individual contrasts of math and language abilities predict STEM disparities. In a longitudinal study of twelfth grade students, those with high ability in both math and language (more girls than boys) were less likely to pursue STEM careers than those with high ability in math and moderate ability in language [6]. Although cultural stereotypes can still be detrimental to girls insofar as they elicit stereotype-threat effects [65] or signal lack of belonging [66], our results imply that girls hold positive associations about their gender group across both math and language ability, consistent with models that depict girls as having more choices in their pursuits, rather than being bound by real or perceived ability constraints. Research that distinguishes between math and language implicit beliefs, then, is important because it can lead to different conclusions about the type of interventions that might be effective for reducing STEM disparities.

## Youths' explicit reports of gender differences in language and math abilities

According to social identity theory [7], in-group preferences frequently emerge when individuals identify with a social category such as gender, and young children (as compared to older children and adolescents) are particularly likely to show in-group preferences [8,9]. We hypothesized that elementary school-aged youth would demonstrate such in-group preferences on explicit measures. On the other hand, because adolescents are more likely than younger children to be aware of conflicting information such as cultural stereotypes favoring boys in math, gender differences in school performance favoring girls, and campaigns in recent decades to promote girls' math engagement, we did not have specific predictions about their explicit beliefs regarding math. In regard to language ability, we predicted that boys and girls would favor girls over boys because cultural stereotypes, gender differences in school performance, and the social acceptability of the language-female stereotype are all congruent. Youths' explicit reports were generally consistent with those hypotheses: *Elementary*-school

aged girls favored girls in both domains, whereas boys reported egalitarian beliefs regarding both language and math. Because of the consistently better average performance of girls in elementary school, these egalitarian reports of boys may represent a lack of calibration to actual achievement disparities and, thus, might be considered as a type of in-group preference. In contrast, for the two older age groups, both boys and girls reported explicit stereotypes favoring girls in math and language abilities.

These explicit reports of adolescents reflect to some extent youth's performance on U.S. national standardized tests. On the National Assessment of Educational Progress (NAEP) exams, girls outperform boys in reading at every grade level, and gender differences in math disappeared in 1996 [67]. Adolescents' beliefs about girls' superiority in math ability might reflect both cultural impetus to encourage girls to take math courses and women to consider math-related careers as well as historical change in the number of women pursuing math, engineering, and computer science degrees. However, although the number of women in math-related fields has increased substantially in the last generation, some gender differences persist. In high school, boys are more likely than girls to take Advanced Placement exams in BC Calculus and Physics [68]. Although women are equally as likely as men to major in math in college, in 2017 approximately twice as many doctoral degrees were awarded to men as to women in the physical and earth sciences, mathematics, and computer sciences, and almost three times as many in engineering [69]. Our results are consistent with the view that although we have not yet reached gender parity in math-intensive fields, the historical landscape of gender differences in math may be changing.

## Relations between implicit and explicit stereotypes

Important contributions of this study were the age comparisons and also the examination of relations between implicit biases and explicit beliefs. Though we had expected to find weak positive relations, especially for elementary-school youth, we did not find any significant correlations between implicit biases and explicit beliefs across any age or gender groups.

These results differ from the meta-analytic findings regarding implicit-explicit correlations in the adult literature [43]. Hofmann and colleagues [43] found that, although general implicit-explicit representations are associated about 0.24 on average, the correlation is lower for stereotypes. They also found that, in adults, a critical moderator of implicit-explicit correlations is the spontaneity of explicit reports. When people relied more on "gut" reactions to report their beliefs, there was a greater congruence between implicit and explicit scores. The lack of significant correlations between math and language implicit gender biases and explicit gender stereotypes in our sample suggests that children as young as 8 years old are not reporting their "gut reactions" or automatic associations to explicit questions about gender differences. Rather, they are controlling their explicit responses. It is also possible that the lack of correspondence in our study might be linked to our choice of measures, and that a different methodological approach might yield significant relations between children's implicit associations and their explicit reports.

## Limitations and recommendations for future research

A significant limitation of the current study was that we had insufficient sample size to explore our research questions within racial/ethnic subgroups. Prior research has shown that gender stereotypes about academic abilities and students' responses to stereotype threat vary according to racial/ethnic identity [18,70,71]. However, we were unable to test whether explicit and implicit beliefs differed across groups due to the lack of racial/ethnic diversity within our sample. Because of the cultural specificity of many gender stereotypes, including academic stereotypes [72], research using either a sample that is homogeneous or samples in which racial,

ethnic, or national groups are compared might further advance our understanding of children and adolescents' academic stereotypes.

A second limitation of the study is that we did not evaluate the personal relevance of the math and verbal domains, and therefore an important caveat to our conclusions is that it is unclear to what extent responses indicate a personal connection to the academic domain versus perceptions of gender differences in ability. For example, although several studies have found that young children tend to show own-gender preferences in competence reports, Cvencek et al. [45] found that as early as second grade, boys showed stronger "me-math" associations than girls. Our measures involved judgements of the gender group in general rather than the individual child's connection to that domain. Girls' implicit bias favoring girls in math in the current study in spite of gender differences at the national level in selections of high school math course-taking would indicate that the implicit scores are a stronger reflection of perceptions of gender-group competence rather than individual identification with the domain. Future research should examine the degree to which gender group competence associations differ from gender differences in individual identification, and age differences in those effects.

A third limitation of the study is that we assessed students' implicit associations and explicit beliefs regarding gender differences in the abilities of youth targets, but not adults. Some studies have shown that children apply cultural academic stereotypes to adults more readily than they do to children [e.g., 33]. Although the explicit reports of adolescents in our sample favored girls in math, youth may have favored men over women had we used adult targets. Such results would reflect gender differences in career choices that favor men in STEM domains and would also be consistent with stereotype threat effects that show performance decrements for women when gender identity is made salient in test situations.

Finally, we note that we did not ask participants whether they were familiar with the Chinese symbols used as neutral stimuli. Although familiarity was unlikely in our sample due to the demographics and location, the effects reported here may be underestimates due to weakened priming effects.

Despite these limitations, this study advances our understanding of youth's implicit biases and explicit beliefs regarding gender differences in academic abilities. We found no correlations between implicit biases and explicit reports, suggesting that youth as young as 8 years old are controlling their responses on explicit measures instead of reporting their automatic associations. Analyses of youth's explicit reports suggested that youth are using information about academic performance and gender stereotype knowledge to adjust their responses, especially as they age. A major goal of our study was to analyze youth's implicit reports across several age groups. Youth's implicit biases were consistent with national gender differences in academic performance, especially for girls. These findings suggest that girls across a wide age range automatically associate good math performance with girls, rather than boys. A key takeaway of both our explicit and implicit findings is that girls strongly favor their in-group in both math and language. These beliefs and positive associations regarding girls' language abilities may, in part, contribute to STEM disparities by giving girls more choices than boys, who perceive their gender group as less-qualified in non-STEM domains.

## Supporting information

**S1 File.**
(PDF)

## Author Contributions

**Conceptualization:** Heidi A. Vuletich, Beth Kurtz-Costes, B. Keith Payne.

**Data curation:** Heidi A. Vuletich, Beth Kurtz-Costes, Erin Cooley.

**Formal analysis:** Heidi A. Vuletich, Beth Kurtz-Costes, B. Keith Payne.

**Funding acquisition:** Beth Kurtz-Costes, B. Keith Payne.

**Investigation:** Heidi A. Vuletich, Beth Kurtz-Costes, Erin Cooley, B. Keith Payne.

**Methodology:** Beth Kurtz-Costes, Erin Cooley, B. Keith Payne.

**Project administration:** Beth Kurtz-Costes.

**Resources:** Beth Kurtz-Costes, B. Keith Payne.

**Software:** B. Keith Payne.

**Supervision:** Beth Kurtz-Costes, B. Keith Payne.

**Writing – original draft:** Heidi A. Vuletich, Beth Kurtz-Costes, B. Keith Payne.

**Writing – review & editing:** Heidi A. Vuletich, Beth Kurtz-Costes, Erin Cooley, B. Keith Payne.

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
