## [Decision Letter · Decision Letter 0]

25 Nov 2019

PONE-D-19-26304

Math and language gender stereotypes: Age and gender differences in implicit biases and explicit beliefs

PLOS ONE

Dear Mrs. Vuletich,

Thank you for submitting your manuscript to PLOS ONE. I had the benefit of receiving feedback from two experts in the field.  I have also had the opportunity to thoroughly consider your paper myself.  As you will see, both reviewers saw a great deal of merit in this work, and I certainly agree with this assessment.  Developing an implicit measure of math-gender stereotyping that can disentangle the potential influences of a competing category (reading/language, etc) is important.  However, the reviewers also raised a number of concerns. I had some related and additional questions while reading this paper and am not yet certain whether they can be adequately addressed through a revision.  As such, after careful consideration, I would like to invite you to submit a revised version of the manuscript that addresses the points raised during the review process so that I can better assess this paper's suitability for publication in PLOS ONE. 

Should you decide to embark on this revision, I will most likely send this paper back out for a second round of reviews and cannot guarantee that it will be accepted following these revisions.  However, in a field where evidence of bias and stereotyping can be more likely to be accepted for publication than evidence that there is no stereotyping, I believe that it is important for the field to be open to publishing these findings.  This is also a high-powered study examining important questions with children and I believe this has the potential to make an important contribution to this literature and to inspire new research.     

I will not reiterate the reviewers' points, but instead will note some of my own: 

My main question had to do with this measure (see also Review 1, point 4).  I found that this stereotyping measure was quite cleverly designed.  It did not initially map onto my vision for this type of AMP. I therefore found myself wondering whether it has previously been validated with adults?  If so, this should be made clear in the introduction.  If not, I wonder what evidence there is that children could complete this measure successfully and that they believed the cover story.  At the risk of seeming self-serving, you might consider referencing previous research that has used the AMP with children to provide some initial evidence (see Perszyk et al., 2019; Williams et al. 2016; Williams & Steele, 2019).  However, these papers either validate the AMP or make use of a child-friendly AMP to examine racial bias.  This stereotyping measure is different in many ways.  For one, I would guess that some children might question whether a language can really have roughly 20 words for good at math, 20 words for bad at math, 20 for good at language arts and 20 for bad at language arts when our own language has no single word to describe these?  Were there any questions to assess the believability of this measure and/or did your instructions make it clear what language the symbols were from (perhaps multiple languages?)?  I agree with Reviewer 2 that the additional blocks might have influenced the effects and/or might provide additional insights. In addition, in my own work using the AMP with children, we found that even with extensive instructions and reminders, a portion of our child participants needed to be removed either because of patterned responding (e.g., the same key or alternating keys on each trial), or because when they were questioned at the end of the measure they reported judging the primes and not the neutral stimuli (despite repeated reminders and extensive instructions/explanations that we wanted them to rate the neutral stimuli).  I noticed that you had excluded very few participants – were there any checks of this sort?  (see also Reviewer 1, minor point 1 regarding exclusions). One concern, of course, is that this is instead some of type of explicit measures, at least for some portion of your child participants.  Another concern, particularly given the use of "good" or "bad" is that this a measure of gender bias (see Baron et al. for related findings of gender bias in children using the IAT).I also found myself having questions about the analyses.  First, why is race treated as a covariate?  What happens if it is not included as a covariate?  Second, I wondered why both math and arts were not included in the same model.  This, of course, would make for a more complex design (a 2x3x2x2 design), but it seems more appropriate.  In fact, part of me wondered whether separating one of the other between subjects variables would make more sense than separating out this within-subjects measure in order to simplify the design.  For example, looking at each age group separately or each gender group separately.  Ideally, of course, this would have all been decided a priori, but looking at your data it would seem that one might draw different conclusions depending on how these data are analyzed.One additional concern, that is noted by both reviewers (point 1 for each), is that I am not quite sure what to make the results.  If the measures are sound, then these results contribute to what are already mixed findings in childhood regarding the development of a math-gender stereotypes.  I would like to see their comments addressed and specific points in the discussion toned down.  For example, I do not agree that, based on your data, the results “are consistent with the idea that implicit linkages between girls and language domains were driving those earlier findings.”       Related to my previous point, I felt that the introduction was not as comprehensive as it might have been.  There are papers that provide evidence that the math-gender IAT predicts math-related outcomes.  These are not consistent with the notion that these biases are driven by a language-girls association.  This should be acknowledged.  I also found the comment in the discussion that “very few studies have examined both implicit biases and explicit beliefs within the same sample” to be inaccurate, as most do.  What they might not do, is report the correlation between the measures

I hope that you will find these and the reviewers' comments to be helpful as you look to revise your manuscript.  Should you decide to resubmit, we would appreciate receiving your revised manuscript by Jan 09 2020 11:59PM. As this is right after the holidays, if you feel that you require more time, please feel free to request it. To enhance the reproducibility of your results, we recommend that if applicable you deposit your laboratory protocols in protocols.io, where a protocol can be assigned its own identifier (DOI) such that it can be cited independently in the future. For instructions see: http://journals.plos.org/plosone/s/submission-guidelines#loc-laboratory-protocols

We look forward to receiving your revised manuscript.

Kind regards,

Jennifer Steele

Academic Editor

PLOS ONE

Journal Requirements:

2. Please provide the full name of the Institutional Review Board that approved your study.

Additional Editor Comments (if provided):

Reviewers' comments:

Reviewer's Responses to Questions

**Comments to the Author**

1. Is the manuscript technically sound, and do the data support the conclusions?

Reviewer #1: Partly

Reviewer #2: Partly

2. Has the statistical analysis been performed appropriately and rigorously? 

Reviewer #1: Yes

Reviewer #2: Yes

3. Have the authors made all data underlying the findings in their manuscript fully available?

Reviewer #1: Yes

Reviewer #2: Yes

4. Is the manuscript presented in an intelligible fashion and written in standard English?

Reviewer #1: Yes

Reviewer #2: Yes

5. Review Comments to the Author

Reviewer #1: This study examined the development of implicit and explicit math-gender stereotypes in 8 to 15 year olds. Results indicate implicitly girl participants favour girls (girls better at math and language vs boys) whereas there is no evidence of preference for boy participants. Explicitly, elementary girls rated girls more highly and elementary boys rated boys more highly in math ability, ratings decreased with age. For explicit language ability, elementary girls (but not boys), middle school boys and girls, and highschool boys and girls rated girls targets as having higher ability than boy targets.

Overall, this study adds to the emerging body of literature focused on the expression of math-gender stereotypes across childhood. On the whole, this literature presents conflicting pattern of results and I don’t believe the current study adds much clarity here, particularly with the explanation of the explicit results. However, this study does make a novel contribution in methodology as the AMP (not the IAT) was used to assess implicit stereotypes. This allows for math and language stereotypes to be examined separately; although gender is still confounded (i.e., stereotypes for girl targets vs boy targets are compared in the two domains). I have raised points below that might be useful to address

1. In the introduction, the argument that social identify theory and ingroup gender preferences drive children’s responses fits for published literature on girl participants (i.e., girls rated as better at academics than boys) but not boys (who also seem to rate girls as better at academics than boys). Is this the best theory to use to explain the whole pattern of results? The authors do a more comprehensive job in the discussion at explaining why the pattern of results might reflect ingroup preference for boys as well. Perhaps this should be worked into the intro as well?

2. A summary of the overall pattern of results for literature on explicit and implicit stereotypes would be helpful. Overall, I get the sense that the results are not consistent. Do the results of the current study aim to clarify the field in any way?

3. I think the paper would benefit from greater clarity regarding for whom the stereotype applies. Research (e.g., Steele, 2003) suggests that children are more likely to apply math-gender stereotypes to adult (but not child) targets. It seems that the literature reviewed is focused on child targets (perhaps explaining the discrepancies in results from different studies). If cultural stereotypes are more readily applied to adults, this has implications for the hypotheses specified page 10 (line 212) as child targets were included in the AMP, and for conclusions regarding the explicit measure.

4. Has the AMP been used to measure stereotypes previously? My concern (especially with the younger children) is that the valance of the response categories trumped the stereotype component. Can this be disentatngled?

5. The implicit and explicit results have a similar pattern in that girls are deemed better in language than boys. Why is this said to reflect “cultural knowledge” for the explicit results but “academic success” for the implicit results? Is it possible that implicit and explicit measures reflect the same underlying constructs, but differences in measurement variability prevent strong correlations from emerging?

On a related point, Page 9, line 186. “Children, however, may not yet have learned the cultural stereotypes and so may vary in awareness”. Or it could be that children have learned the stereotype, but have not yet internalized it to the point it can be automatically activated by attitude object. What are the implications of these possibilities for the hypotheses and results?

Minor Points

1. 2.2% of sample of Asian heritage. Did these participants have any familiarity with Chinese symbols. If so, should they be removed?

2. What were the correlations for the four prime-gender scores? (page 20)

3. Tone down language around conclusions. For example, p 21 line 328; p 23, line 345. Other studies (using the IAT) have demonstrated age-related differences in implicit biases.

Reviewer #2: This article explored the development of gender differences in stereotypes about math and language. Children were administered an implicit and explicit gender stereotype measure across age groups. The manuscript aims to tackle an important issue - distinguishing (in measurement) stereotypes about math from stereotypes about language which, with the exception of a few studies, has not been investigated much.

The study reports that girls have an implicit own group gender bias (thinking own gender is better at math and language) whereas boys do not have an implicit gender bias for either domain.

There are several areas that I think could benefit from revision.

1. It would help if the authors could make greater sense of the implicit data from boys. That is, these findings seem to contradict past published work where boys show an implicit gender stereotype. Is there a way to compare the strength of the egalitarian associations with math and language to see if the effect is stronger in one direction? There are now a number of papers by Cvencek, as well as those who have done stereotype threat work (Tomasetto, Steele etc) arguing that in some way shape or form boys have a gender stereotype in this domain. Is there something unique about how the IAT measures bias that might make it a more suitable measure in this case? Of course, the data are what they are but I think much more attention should be given to this contradictory finding both in terms of possible methodological explanations as well conceptual. Related, can the authors report more info on average latencies with the AMP? It might help to understand how implicit these responses likely were.

2. I didn't follow the arguments the authors made about how the data on the implicit/explicit measures directly speaks to the sources of these stereotypes. That is, to say that if implicit bias is more influenced by cultural messages about stereotypes then they should increase with age doesn't make clear sense to me. And, by contrast, classroom cue sensitivity would lead to no age differences (lines 210-214). First, it's odd that there wasn't a direct measure of sensitivity to cultural stereotypes or some quantification of classroom cues. It was assumed that patterns of bias uniquely are constrained by these cues when there are a multitude of factors that also uniquely shape bias (e.g., surely they interact). Further, it is not clear that being influenced by cultural messages about stereotypes should mean that the bias increases with age (there isn't strong evidence that implicit bias reflects a cumulative learning model whereby the magnitude of the bias increases with frequency of exposure), there could be sensitive periods for learning biases, etc. Similarly, what's the evidence the present classroom made available diagnostic cues to performance/ability?

3. The authors noted that other school domains were studied but not the focus of the present manuscript. Rarely do I think these additional study data are informative but for the present manuscript I especially think it's informative because it speaks to broader arguments the paper seems to be very focused on - own group bias, internalization/awareness of cultural stereotypes and classroom cues. Do any of the other data not reported shed light on children's more general sensitivities here?

4. I think it would be helpful if the authors could include more discussion on the growing stereotype threat literature in this domain as it seems to be particularly informative for our thinking and predictions about the development of gender differences in these academic stereotypes. And, in some cases, may even present contradictory findings that require some explanation.

5. The authors setup two primary views about measuring bias - importance and limitations for studying both explicit and then implicit bias. This made sense. I got lost a bit when indirect measures were then discussed because, conceptually, I didn't understand where the authors saw indirect measures fitting in the literature - is it a level of analysis like implicit/explicit or is it kinda orthogonal to the implicit/explicit distinction and more of a way to measure things explicitly while reducing some the potential demand characteristics that can plague explicit measures?

6. Lastly, can the authors highlight/note the analyses they were likely underpowered for given then power analysis they did as the effect sizes reported in a number of cases seemed to be below the threshold they set for the study.

I am excited and inspired by this work as it is important theoretically and methodologically to be examining this issues. Thanks!

6. PLOS authors have the option to publish the peer review history of their article (what does this mean?). If published, this will include your full peer review and any attached files.

Reviewer #1: No

Reviewer #2: No

---

## [Author Response · Author response to Decision Letter 0]

31 Mar 2020

Editor’s Comment #1: My main question had to do with this measure (see also Review 1, point 4). I found that this stereotyping measure was quite cleverly designed. It did not initially map onto my vision for this type of AMP. I therefore found myself wondering whether it has previously been validated with adults? If so, this should be made clear in the introduction. If not, I wonder what evidence there is that children could complete this measure successfully and that they believed the cover story. At the risk of seeming self-serving, you might consider referencing previous research that has used the AMP with children to provide some initial evidence (see Perszyk et al., 2019; Williams et al. 2016; Williams & Steele, 2019). However, these papers either validate the AMP or make use of a child-friendly AMP to examine racial bias. This stereotyping measure is different in many ways. For one, I would guess that some children might question whether a language can really have roughly 20 words for good at math, 20 words for bad at math, 20 for good at language arts and 20 for bad at language arts when our own language has no single word to describe these? Were there any questions to assess the believability of this measure and/or did your instructions make it clear what language the symbols were from (perhaps multiple languages?)? I agree with Reviewer 2 that the additional blocks might have influenced the effects and/or might provide additional insights.

Response: This version of the AMP, focusing specifically on academic stereotypes, has not been validated in adults or children, though we did conduct a pilot study to determine the most appropriate presentation time for each stimulus. We have clarified the point about validity in the introduction and also added the suggested references discussing previous use of the AMP with children (bottom of pg. 13). Thank you for this suggestion. 

We did not assess the believability of the measure, in part because research with adults has shown that people have difficulty introspecting on how they judged the stimuli and tend to confabulate reasons (Payne et al., 2013). Moreover, the instructions were ambiguous as to whether the symbols actually meant “good at math” or “bad at math” (same for language arts). Our instructions stated that we were interested in how people make guesses about the meaning of words, and that they should guess that about half the symbols meant “good at math” and about half meant “bad at math.” We said that they should use their “feelings” to make a guess. We are now more explicit about the wording of these instructions in the Method section (pg. 19). 

Regardless of children’s interpretation or suspicion of the instructions, the systematic differences that we found between boys’ and girls’ ratings of different gendered primes were inconsistent with their controlled responses as measured by the explicit measure. Yet, these systematic differences suggest that children did not respond completely at random either. One possibility, as raised by the editor and one of the reviewers, is that children ignored the academic subject and simply responded with their gender biases of girls versus boys as “good” or “bad.” To test this possibility, we ran additional analyses examining children’s sports implicit biases. As with math and language implicit biases, we found a significant gender x prime gender interaction. However, the pattern of results was different compared to the academic biases. For sports, boys showed an implicit bias favoring boys in sports, whereas girls did not show a gendered bias. As a reminder, for math and language, girls had shown an implicit bias favoring girls and boys had not shown a bias favoring either gender. These results suggest that the AMP was measuring domain-specific implicit biases rather than general gender biases. These results are now included in the supplemental information (and briefly mentioned in the main manuscript, bottom of pg. 24). 

Editor’s Comment #2: In addition, in my own work using the AMP with children, we found that even with extensive instructions and reminders, a portion of our child participants needed to be removed either because of patterned responding (e.g., the same key or alternating keys on each trial), or because when they were questioned at the end of the measure they reported judging the primes and not the neutral stimuli (despite repeated reminders and extensive instructions/explanations that we wanted them to rate the neutral stimuli). I noticed that you had excluded very few participants – were there any checks of this sort? (see also Reviewer 1, minor point 1 regarding exclusions). One concern, of course, is that this is instead some of type of explicit measures, at least for some portion of your child participants. Another concern, particularly given the use of "good" or "bad" is that this a measure of gender bias (see Baron et al. for related findings of gender bias in children using the IAT).

Response: We have now included a statement clarifying that none of our participants met our exclusion criterion, which was pressing the same key on all trials (pg. 20). As for pressing alternating keys on each trial, we are not able to identify this type of patterned responding because the stimuli were randomly ordered for each participant and during each block. The software that we used did not record the order of stimuli presentation for each participant. 

Please see our response to comment #1 above regarding the possibility that the AMP, in this context, was a type of explicit measure. First, children’s math and language implicit biases were not correlated with their explicit reports. Second, their implicit biases in sports showed a different pattern of bias by gender group, suggesting that the measure was sensitive to domain. Third, the implicit sport biases of children showed a small, but significant, correlation with their explicit beliefs about sports competence (r = .15). These results suggest that this version of the AMP was measuring systematic differences in automatic associations to academic subjects based on gender categories, and that for less socially sensitive stereotypes (like those about sports) there is a small correlation between implicit and explicit measures. We now include these robustness checks in the supplemental information. Thank you for raising this important point and encouraging us to run additional analyses. 

We also included a new section in the introduction dedicated to discussing the AMP in detail (pg. 12).

Editor’s Comment #3: I also found myself having questions about the analyses. First, why is race treated as a covariate? What happens if it is not included as a covariate? Second, I wondered why both math and arts were not included in the same model. This, of course, would make for a more complex design (a 2x3x2x2 design), but it seems more appropriate. In fact, part of me wondered whether separating one of the other between subjects variables would make more sense than separating out this within-subjects measure in order to simplify the design. For example, looking at each age group separately or each gender group separately. Ideally, of course, this would have all been decided a priori, but looking at your data it would seem that one might draw different conclusions depending on how these data are analyzed.

Response: We are happy to clarify these points. Race was treated as a covariate because academic stereotypes can sometimes differ by race categories. Unfortunately, we did not have adequate numbers of youth in each racial group to test race differences, so we controlled for race instead. The results remain the same when race is removed from the analyses, except that the explicit three-way academic subject x target gender x grade interaction was reduced to a significant academic subject x prime gender interaction where children across all three grades (not just elementary school) favored girls in math. We now leave out race as a covariate. 

We initially chose to analyze academic subjects in separate models to simplify the analyses, avoiding the possibility of a four-way interaction. However, we appreciate the editor’s point that it is more appropriate to keep this within-subject variable in one model. We now report our results using a 2x3x2x2 design. The findings remain the same. 

Editor’s Comment #4: One additional concern, that is noted by both reviewers (point 1 for each), is that I am not quite sure what to make the results. If the measures are sound, then these results contribute to what are already mixed findings in childhood regarding the development of a math-gender stereotypes. I would like to see their comments addressed and specific points in the discussion toned down. For example, I do not agree that, based on your data, the results “are consistent with the idea that implicit linkages between girls and language domains were driving those earlier findings.” 

Response: Thank you for raising this concern. We have now clarified our findings’ theoretical contributions. First, we clarify that the vast majority of previous studies examining math and language implicit biases in children or adolescents have used the IAT (pg. 8). These studies show a clear pattern of results among girls of all age-groups: girls show a male-math bias and this bias may predict math-related outcomes such as enrollment intentions for math courses and math self-concepts. However, because of the nature of paired categories in the IAT, these results could be due to girls having a strong association of math with boys, a strong association of girls with language, or both. We found only two studies that used implicit measures that did not confound math and language associations. In one, girls demonstrated an own-gender bias across all grades (4-6). In the other, 9th grade girls did not show a bias for either gender. In other words, the math-male bias among girls may only appear when math and language gender associations are confounded. The implication is that girls may not, in fact, have a negative association of their own gender to math. Rather, when math and language are directly contrasted, they might preferentially associate their own gender with language. Our study is important because it adds evidence that girls have an in-group math bias as well as an in-group language bias (summary at the end of pg. 11). Our findings highlight the need for more research that uses implicit measures that do not confound math and language biases to understand if and for what groups these findings replicate. 

Editor’s Comment #5: Related to my previous point, I felt that the introduction was not as comprehensive as it might have been. There are papers that provide evidence that the math-gender IAT predicts math-related outcomes. These are not consistent with the notion that these biases are driven by a language-girls association. This should be acknowledged. I also found the comment in the discussion that “very few studies have examined both implicit biases and explicit beliefs within the same sample” to be inaccurate, as most do. What they might not do, is report the correlation between the measures. 

Response: We have now expanded the introduction to describe previous explicit and implicit findings in more detail, and in particular, findings that link math-gender IAT to math outcomes (bottom of pg. 9). Likewise, in the general discussion, we expand upon why we believe that strong language-girls associations (as compared to math-girls associations), might still predict academic outcomes in math-related domains. In page 33, we added: 

Though, at first glance, our results might not appear consistent with gender disparities in STEM careers, they are revealing in that they support recent theoretical frameworks suggesting that girls opt out of math, not due to perceived deficit in math ability compared to boys, but due to perceived strength in language ability over math ability. For example, a large international study of 15-year-old students found that girls’ comparative advantage in reading as opposed to math can largely explain gender disparities in intentions to pursue math-related careers (67). In that study, girls who were found to be good at math were more likely than boys to be even better at reading than at math. The gap between math and reading performance accounted completely for gender differences in math self-concept, interest in math, and attitudes towards math. Other studies have also found that intra-individual contrasts of math and language abilities predict STEM disparities. In a longitudinal study of twelfth grade students, those with high ability in both math and language (more girls than boys) were less likely to pursue STEM careers than those with high ability in math and moderate ability in language (6). Although cultural stereotypes can still be detrimental to girls insofar as they elicit stereotype-threat effects (68) or signal lack of belonging (69), our results imply that girls hold positive associations about their gender group across both math and language ability, consistent with models that depict girls as having more choices in their pursuits, rather than being bound by real or perceived ability constraints. Research that distinguishes between math and language implicit beliefs, then, is important because it can lead to different conclusions about the type of interventions that might be effective for reducing STEM disparities.

We have also deleted our comment about other studies not measuring implicit biases and explicit beliefs within the same sample.

Reviewer 1

Reviewer 1, Comment #1: In the introduction, the argument that social identify theory and ingroup gender preferences drive children’s responses fits for published literature on girl participants (i.e., girls rated as better at academics than boys) but not boys (who also seem to rate girls as better at academics than boys). Is this the best theory to use to explain the whole pattern of results? The authors do a more comprehensive job in the discussion at explaining why the pattern of results might reflect ingroup preference for boys as well. Perhaps this should be worked into the intro as well?

Response: We have expanded and restructured much of the introduction to make the pattern of results (or lack thereof) for past studies clearer. We also revised the way we discuss specific theories for explaining children’s responses on explicit measures. More specifically, we acknowledge different theoretical perspectives (including social identity theory), but point out that published research on this topic has yielded inconsistent findings (pg.6-7). Our view is that multiple factors likely influence explicit reports due to participants’ ability to control their responses. This challenge with explicit measures highlights the advantage of using implicit bias measures.

Reviewer 1, Comment #2: A summary of the overall pattern of results for literature on explicit and implicit stereotypes would be helpful. Overall, I get the sense that the results are not consistent. Do the results of the current study aim to clarify the field in any way?

Response: Thank you for this suggestion. We have included summary statements for the explicit (pg. 7, line 138) and implicit findings (pg. 11, line 238). For explicit findings, there is no overall pattern. We clarify this point and suggest reasons for why this might be true. For implicit findings, the pattern is very consistent for studies using IAT measures; youth (girls in particular) show a stereotypical math-male bias and language-female bias. Studies that did not rely on IAT, but used measures that disambiguate math/language biases, did not find math-male biases among girls. Our study aims to reconcile these existing, conflicting findings by providing evidence that measurement differences could account for the seemingly inconsistent findings in the implicit bias literature (specific to math/language biases). We find that girls show a counter-stereotypical math-female bias. These findings are important for our understanding of how automatic associations about math and language ability influence motivation and career trajectories. 

Reviewer 1, Comment #3: I think the paper would benefit from greater clarity regarding for whom the stereotype applies. Research (e.g., Steele, 2003) suggests that children are more likely to apply math-gender stereotypes to adult (but not child) targets. It seems that the literature reviewed is focused on child targets (perhaps explaining the discrepancies in results from different studies). If cultural stereotypes are more readily applied to adults, this has implications for the hypotheses specified page 10 (line 212) as child targets were included in the AMP, and for conclusions regarding the explicit measure.

Response: Thank you for pointing out this important distinction (i.e., to what age group the stereotype applies). We chose to focus on child targets for two reasons. First, as you pointed out, much of the prior literature examining youths’ academic gender stereotypes has used child targets. Second, youths’ views of the competence of social group members—in this case, their perceptions of the abilities of boys and girls who are roughly their age—are known to be related to their perceptions of their own abilities, their domain-specific interests, and other motivational variables. We have added this point as a study limitation in the Discussion as excerpted below (pg. 35): 

A third limitation of the study is that we assessed students’ implicit associations and explicit beliefs regarding gender differences in the abilities of youth targets, but not adults. Some studies have shown that children apply cultural academic stereotypes to adults more readily than they do to children (e.g., 33). Although the explicit reports of adolescents in our sample favored girls in math, youth may have favored men over women had we used adult targets. Such results would reflect gender differences in career choices that favor men in STEM domains and would also be consistent with stereotype threat effects that show performance decrements for women when gender identity is made salient in test situations.

Reviewer 1, Comment #4: Has the AMP been used to measure stereotypes previously? My concern (especially with the younger children) is that the valance of the response categories trumped the stereotype component. Can this be disentangled?

Response: This is an excellent point. We now include supplemental analyses showing that boys favor boys in sports, rather than girls, whereas girls show no bias. These results suggest that the AMP was testing domain-specific gender associations. Please see our response to Editor’s Comment #1. 

Reviewer 1, Comment #5: The implicit and explicit results have a similar pattern in that girls are deemed better in language than boys. Why is this said to reflect “cultural knowledge” for the explicit results but “academic success” for the implicit results? Is it possible that implicit and explicit measures reflect the same underlying constructs, but differences in measurement variability prevent strong correlations from emerging?

On a related point, Page 9, line 186. “Children, however, may not yet have learned the cultural stereotypes and so may vary in awareness”. Or it could be that children have learned the stereotype, but have not yet internalized it to the point it can be automatically activated by attitude object. What are the implications of these possibilities for the hypotheses and results?

Response: We agree with the reviewer that our results do not speak to the specific constructs underlying the observed biases. We changed the language in our discussion to reflect their speculative nature. We also added a sentence in the Discussion regarding the possibility that measurement might have led to the lack of correlation between implicit and explicit stereotypes as excerpted below:

 It is also possible that the lack of correspondence in our study might be linked to our choice of measures, and that a different methodological approach might yield significant relations between children’s implicit associations and their explicit reports (p. 34, line 674).

Reviewer 1, Comment #6: 2.2% of sample of Asian heritage. Did these participants have any familiarity with Chinese symbols. If so, should they be removed?

Response: Only six participants reported their race as Asian-American. If we exclude these participants, the results remain exactly the same; therefore, we retained their observations. 

Reviewer 1, Comment #7: What were the correlations for the four prime-gender scores? (page 20)

Response: We now report these correlations in Table 7. 

Reviewer 1, Comment #8: Tone down language around conclusions. For example, p 21 line 328; p 23, line 345. Other studies (using the IAT) have demonstrated age-related differences in implicit biases.

Response: We have made these suggested changes. 

 

Reviewer 2

Reviewer 2, Comment #1: It would help if the authors could make greater sense of the implicit data from boys. That is, these findings seem to contradict past published work where boys show an implicit gender stereotype. Is there a way to compare the strength of the egalitarian associations with math and language to see if the effect is stronger in one direction? There are now a number of papers by Cvencek, as well as those who have done stereotype threat work (Tomasetto, Steele etc) arguing that in some way shape or form boys have a gender stereotype in this domain. Is there something unique about how the IAT measures bias that might make it a more suitable measure in this case? Of course, the data are what they are but I think much more attention should be given to this contradictory finding both in terms of possible methodological explanations as well conceptual. Related, can the authors report more info on average latencies with the AMP? It might help to understand how implicit these responses likely were.

Response: We now clarify that several past studies have also failed to find stereotypic implicit biases in boys. In fact, the findings among boys are much more inconsistent than those among girls. That said, we have also incorporated the references suggested above which document gender stereotypes among boys and have worked to clarify that we are unsure about why boys and girls show dissociative implicit biases (pg. 12):

We were agnostic about the implicit biases of boys, as previous findings have been inconsistent and do not clearly favor one theoretical account over another. The dissociative processes by which girls and boys form automatic associations is in itself interesting, but not the subject of this report.

We still maintain the following: 

The view of academic success as a feminine trait may have led girls in the present study to show implicit biases favoring girls in both domains, whereas for boys, those views may have been tempered by a tendency to show in-group preference, resulting in their egalitarian scores on the task. (bottom of pg. 32)

As for the associations being stronger in the direction of math or language, we can now speak to that question. We changed our analyses such that math and language are included in the same model (a 2x3x2x2 design). The three-way interaction (i.e., academic subject x prime gender x gender) that might reveal differences in the relative strength of math versus language associations among boys was not significant. Finally, we cannot report information on average latencies because the AMP does not measure reaction times. Rather, it measures the proportion of targets that the participants rate as “good at” or “bad at” math/language when preceded by a photo of either a boy or a girl. 

Reviewer 2, Comment #2: I didn't follow the arguments the authors made about how the data on the implicit/explicit measures directly speaks to the sources of these stereotypes. That is, to say that if implicit bias is more influenced by cultural messages about stereotypes then they should increase with age doesn't make clear sense to me. And, by contrast, classroom cue sensitivity would lead to no age differences (lines 210-214). First, it's odd that there wasn't a direct measure of sensitivity to cultural stereotypes or some quantification of classroom cues. It was assumed that patterns of bias uniquely are constrained by these cues when there are a multitude of factors that also uniquely shape bias (e.g., surely they interact). Further, it is not clear that being influenced by cultural messages about stereotypes should mean that the bias increases with age (there isn't strong evidence that implicit bias reflects a cumulative learning model whereby the magnitude of the bias increases with frequency of exposure), there could be sensitive periods for learning biases, etc. Similarly, what's the evidence the present classroom made available diagnostic cues to performance/ability?

Response: We agree with the reviewer that our results do not speak to the specific sources underlying the observed biases. We changed the language in our introduction and discussion to reflect the fact that our data cannot speak to mechanisms. We are also now explicit in the introduction about the fact that multiple factors can influence beliefs and biases. Thank you for raising this point. 

Reviewer 2, Comment #3: The authors noted that other school domains were studied but not the focus of the present manuscript. Rarely do I think these additional study data are informative but for the present manuscript I especially think it's informative because it speaks to broader arguments the paper seems to be very focused on - own group bias, internalization/awareness of cultural stereotypes and classroom cues. Do any of the other data not reported shed light on children's more general sensitivities here?

Response: Thank you for this suggestion. We now include supplemental analyses showing that boys and girls have different implicit biases in sports compared to math and language, suggesting that our implicit measure was sensitive to both gender and academic domain. Please see our response to Editor’s Comment #1. 

Reviewer 2, Comment #4: I think it would be helpful if the authors could include more discussion on the growing stereotype threat literature in this domain as it seems to be particularly informative for our thinking and predictions about the development of gender differences in these academic stereotypes. And, in some cases, may even present contradictory findings that require some explanation.

Response: This is an excellent point. Because stereotype threat refers to performance decrements in a test situation when membership in a negatively-stereotyped group is activated, results of the current study are not directly related to stereotype threat. Nonetheless, we added that point in the Discussion as excerpted below:

Although cultural stereotypes can still be detrimental to girls insofar as they elicit stereotype-threat effects (68) or signal lack of belonging (69), our results imply that girls hold positive associations about their gender group across both math and language ability, consistent with models that depict girls as having more choices in their pursuits, rather than being bound by real or perceived ability constraints (pg. 33, line 255). 

Reviewer 2, Comment #5: The authors setup two primary views about measuring bias - importance and limitations for studying both explicit and then implicit bias. This made sense. I got lost a bit when indirect measures were then discussed because, conceptually, I didn't understand where the authors saw indirect measures fitting in the literature - is it a level of analysis like implicit/explicit or is it kinda orthogonal to the implicit/explicit distinction and more of a way to measure things explicitly while reducing some the potential demand characteristics that can plague explicit measures?

Response: Thank you for raising this point. We agree that results from indirect measures are not relevant to the current study and have removed reference to these studies. 

Reviewer 2, Comment #6: Lastly, can the authors highlight/note the analyses they were likely underpowered for given then power analysis they did as the effect sizes reported in a number of cases seemed to be below the threshold they set for the study.

Response: We have noted in the Results section the effect that fell below the threshold calculated from our sensitivity analysis. Excerpted from pg 21: 

The Age Group x Academic Subject x Target Gender interaction was also significant, F(2, 258) = 3.51, p = .031, η2 = .03. Children favored girls in language across all three age groups. In contrast, in the case of math, only children in elementary school showed a bias, favoring girls over boys in math. Youth in middle school and high school did not show a gender bias in explicit reports of math ability. These results should be interpreted with caution, though, as the effect size is below the threshold calculated by our sensitivity analysis. The four-way interaction was not significant, perhaps due to low power.

---

## [Decision Letter · Decision Letter 1]

3 Jun 2020

PONE-D-19-26304R1

Math and language gender stereotypes: Age and gender differences in implicit biases and explicit beliefs

PLOS ONE

Dear Dr. Vuletich,

Thank you for submitting this revised manuscript to PLOS ONE. I have had the opportunity to read through your re-submission of this manuscript and have again had the benefit of receiving feedback from the two original reviewers. As you will see in their reviews, as well as my own comments below, we continue to feel that this manuscript has a great deal of promise. I believe that there is tremendous benefit to using a range of measures to gain a deeper understanding of the early developmental of implicit academic stereotypes. I also appreciate the additional information that you have provided in the supporting information document, which strengthens your interpretation of the data. 

You will also see that the reviewers and I also continue to raise some concerns. I believe that these can be addressed in a revision, and therefore would like to invite you to submit a revised version of the manuscript that addresses the points raised below. I cannot guarantee that this revised version will be accepted for publication, but I do not plan to send this back out for another round of reviews prior to making my decision.

I would encourage you to work to address each of the reviewer’s comments, with a focus on additional limitations that will need to be noted in the discussion section. In particular, Reviewer 1 raised two main points that will need to be adequately addressed in the discussion. That is, given the different nature of this particular measure, you cannot conclude that the implicit language-stereotyping effect for girls is driving the math-gender stereotyping on the IAT (more on that below). Reviewer 2 raises a number of important points, many of which should at the very least be addressed in the discussion. In particular, the relative lack of exclusion criteria should be discussed relative to other papers that make use of implicit measures with child participants, with a focus on what might have been done to ensure that your effects are not simply the result of a great deal of noisy participants in the data (more on that from me below as well).

In addition, my own comments include the following:

The inclusion of the implicit sports biases measure was a great addition. But this led me to wonder what other measures were included in this study. I would request that in the current climate of replicability, these be outlined in the supporting information. Is it possible that some effects did not emerge because children were fatigued by the time they were completing the key measures? Could these effects have been influenced by other measures that they completed prior to the main measures of interest?I was unclear what the M*diff *score was for the implicit sports bias in the supporting information.  More information on that measure is needed. I would also recommend that this get integrated into the main manuscript as I believe that the measure has the potential to speak against a very clear alternative explanation. However, please see Reviewer 1 as the limitation to this possibility needs to be fully addressed in the discussion.Another way that the positivity bias could be addressed is by examining responses to your different stimuli. I wonder, if you separate the Black and White targets, do you see a race bias? That is, were participants more likely to select “good at” when presented with a White target prime as opposed to a Black target prime? Is this particularly true for the Black male primes and the non-Black participants? (see Perszyk et al., 2019). This might be worth exploring and might provide additional insight that could be included and considered in the supplement.The paragraph that starts on line 276 is meant to address my concern that many of your participants might have rated the primes instead of the neutral stimuli. However, I did not find this to really address the concern.  As noted by Reviewer 2 and by me above, the possibility exists that the lack of stereotyping is due to a small effect combined with noise in your child data due to a relative lack of exclusion criteria. Please see the other AMP papers with children (Williams et al., Williams & Steele, Perszyk et al.) for the criteria that were used. If you can apply these, wonderful – if not, this limitation needs to be more fully addressed in the discussion. I will also mention that I found the exclusion criteria description on lines 447-448 to be awkward and would recommend rewording.I am conflicted about the analyses that are presented and would suggest that you give some additional thought to them.  In the absence of the four-way interaction, you ultimately do not directly test your hypotheses that, for example, “the youngest age group…(will) favor their own gender in math”, etc. I know that one school of thought is that these direct comparisons should not be made in the absence of the interaction effects. On the other hand, given the complexity of your design and your specific hypotheses, I wondered whether including more direct tests would be helpful in making full sense of these data. I also wondered whether the conclusions that are drawn on lines 584-585 completely align with the analyses that you conduct. I do think that providing access to the data – which I don’t think were submitted, but you note the intention to link to these – might solve this issue. But I would be open to you including some additional analyses in the supplemental materials or main text.   

Some additional suggestions include:

I found ‘road map’ statements to be unnecessary and would recommend that sentences like the one on line 260 be removed.The reliability of this measure is easily calculated and could be included with the measure. Please include it and remove the justification (on page 14) for not including this.What is the theoretical justification for the age grouping that was used?  Please provide some justification.I would report on the implicit measure first in the measures and results and then would report on the explicit measure for consistency.Line 456 – Prime Gender should be Target Gender.  Additional interactions could be fully reported in the supplement.Table 7 – it would be interesting to see this for boys and girls separately as well.

Overall, I think that there are some real strengths to this manuscript, and I believe that it has the potential to make an important contribution to the field.  I hope that you will decide to address each of these concerns and resubmit the paper for additional consideration.

Please submit your revised manuscript by Jul 18 2020 11:59PM. This is the revision date set by the journal, however, if you will need more time than this to complete your revisions this is not a problem. Please reply to this message or contact the journal office at plosone@plos.org. Please include the following items when submitting your revised manuscript:

I hope that you and your co-authors are staying well at this strange and challenging time. I will look forward to receiving your revised manuscript and will aim to render a decision as quickly as possible after it is received.

Warmly,

Jennifer Steele

Academic Editor

PLOS ONE

Reviewers' comments:

Reviewer's Responses to Questions

**Comments to the Author**

1. If the authors have adequately addressed your comments raised in a previous round of review and you feel that this manuscript is now acceptable for publication, you may indicate that here to bypass the “Comments to the Author” section, enter your conflict of interest statement in the “Confidential to Editor” section, and submit your "Accept" recommendation.

Reviewer #1: (No Response)

Reviewer #2: (No Response)

2. Is the manuscript technically sound, and do the data support the conclusions?

Reviewer #1: Partly

Reviewer #2: Partly

3. Has the statistical analysis been performed appropriately and rigorously? 

Reviewer #1: Yes

Reviewer #2: Yes

4. Have the authors made all data underlying the findings in their manuscript fully available?

Reviewer #1: Yes

Reviewer #2: Yes

5. Is the manuscript presented in an intelligible fashion and written in standard English?

Reviewer #1: Yes

Reviewer #2: Yes

6. Review Comments to the Author

Reviewer #1: I believe that this revised manuscript is stronger than the original submission. The comments I raised in my review have been adequately addressed to the extent that the data allows for this. There continues to be many strengths to this paper (i.e., Social ID theory, sample, methods, robust analyses, etc). Most importantly, I agree with the authors that the field would benefit from research that uses diverse implicit measures. The current paper meets that objective without doubt. However, I continue to have two conceptual concerns that may prevent this paper from being publishable in PLOS One.

1. The authors argue that implicit language associations could be driving the IAT math-gender stereotype findings. I agree. My issue is that there is no data to support this claim (pages 32-33, line 612-618). Instead what we see is a more general implicit girls = good pattern of results. I note that there could still be a contribution to make here in that social id theory can be used to explain this pattern of results, but I am uncertain whether this is novel enough for publication in PLOS One.

2. This girls = good issue was raised in the original reviews. To address this the authors examined responses on a Sport-AMP and found that for male participants boys were more positively associated with sport, girls demonstrated no bias (mirroring the academic-related AMPs where boys showed no bias). In my opinion, this analysis does not adequately address the issue. What we may be seeing here is a broader "girls = good at school / boys = good at sports" stereotype that would be consistent with input via cultural exposure. Again, social identity theory could be used for a framework to interpret this pattern of results.

Reviewer #2: I commend the effort to address so many of the reviewer comments in a thoughtful, clear and concise way. There are some issues that I do think are still quite important to tend to as it bears directly on the framing and claims.

The primary focus is to advance our understanding of implicit and explicit gender stereotypes. As such, we want to have some reasonable comfort that the measures are indeed capturing something implicit (and explicit). How can one tell if it's implicit? As we know, there are a variety of different ways to address this, some better than others. But I'm not sure what can be said here as this procedure hasn't been established with children. Not a direct line into what's implicit, an earlier reviewer comment re latency data would be quite useful - particularly if latencies were quite slow. I understand from the response letter that AMPs calculate proportions of response types. Is it the case that the software used really doesn't capture latency data for each trial? I understand the AMP analysis doesn’t incorporate these data but my question is asking whether the software itself has such data. What software was used? Most programs I know capture these data. Assuming this isn’t available, what then can we point to as evidence that this procedure with children has been shown to capture implicit (as opposed to explicit) bias?

How come the presentation stimuli times differed from the one AMP study with children to date?

I remain concerned about not checking for whether participants were familiar with the Chinese characters. Imagine, for example, this were taking place today with the rising amounts of overt racism toward China. I could imagine familiarity with the characters could present two issues. 1. Prime negative affect itself. 2. Lead participants to doubt that the characters actually stand for words meaning good/bad at x,y,z if they have a mutual exclusivity hypothesis about Chine language (one character per concept, similar to word learning bias in English and other Western languages). Perhaps this believability doesn't matter?

As well, I’m still confused by how we can reasonably conclude this is not an attitude measure toward the primed stimuli (vs stereotypes). Is the study powered for the Sports AMP that was used to demonstrate there isn’t just a positivity bias? Is there a correlation between the two AMPS (presumably there would not be if it were not measuring a general gender good/bad bias)? Was there an order effect with the different AMPs conducted? Is there a domain difference for boys/girls (on ave or by gender) to further help us to see if they’re indeed capturing different constructs?

I’m puzzled by the very lax exclusion criteria for the AMP. With IAT, for example, exclusion criteria is around 20% errors or greater. Alternating key presses would be missed and this is not uncommon for children. Before I can really make sense of these data I’d want to see much clearer reporting of proportion of trials with one key press, what the range is, SD, etc for age groups. What software was used to run this program?

Do we think the mixed results for explicit gender stereotypes reported in the intro is conceptual or methodological? That is, does it reflect variability due to differences in personal and or cultural stereotypes represented by the child or due to methodological differences employed or something else? This would be helpful to discuss perhaps somewhere (but doesn't haven't to be solved here).

The authors note “We expected girls would show traditional math-male biases if

they have assimilated cultural stereotypes that favor boys in math. In contrast, girls would favor

girls in math if pervasive differences in academic performance are the primary factor shaping

automatic associations about gender and math ability. “ Is it possible they could hold both stereotypes and randomly (or non-randomly as with some kind of prime) exhibit one of these stereotypes in the moment? Some children were tested in a local library. Were these more female than male? Were stereotype assessments different here (potentially because of its linkage to language- reading). Given sensitivity on explicit measures to social desirability, was there an effect of experimenter gender on the explicit measure for older children?

I was to reiterate how important I think it is to measure stereotypes about math/reading/language separately as this confound is really apparent when thinking about the existing findings in this domain with the IAT.

7. PLOS authors have the option to publish the peer review history of their article (what does this mean?). If published, this will include your full peer review and any attached files.

Reviewer #1: Yes: Amanda Williams

Reviewer #2: No

---

## [Author Response · Author response to Decision Letter 1]

14 Jul 2020

Our response to reviewer and editor comments is attached.

---

## [Editor Report · Decision Letter 2]

13 Aug 2020

Math and language gender stereotypes: Age and gender differences in implicit biases and explicit beliefs

PONE-D-19-26304R2

Dear Dr. Vuletich,

I have now had the opportunity to review your most recent submission of this paper. I feel that you did an excellent job integrating the suggestions made by both me and the reviewers. I am therefore pleased to inform you that your manuscript has been judged scientifically suitable for publication and will be formally accepted for publication once it meets all outstanding technical requirements. I feel confident that this paper will make an important contribution to our field.

I have one final suggestion for you to consider as you finalize the supplement for publication. I noticed that in the S1 Table the pairwise comparisons focus only on explicit language stereotypes. I feel that it would be helpful to have a comparable table containing explicit math stereotypes. This is not a requirement, but rather is a suggestion that could be integrated into the supplement (or posted on the OSF) should you agree.

Within one week, you will receive an e-mail detailing the required amendments. When these have been addressed, you’ll receive a formal acceptance letter and your manuscript will be scheduled for publication.

I want to commend you on this interesting research and I look forward to seeing this paper published in PLOS ONE.

Warmly,

Jenn Steele

Academic Editor

PLOS ONE

---

## [Editor Report · Acceptance letter]

24 Aug 2020

PONE-D-19-26304R2 

Math and language gender stereotypes:Age and gender differences in implicit biases and explicit beliefs 

Dear Dr. Vuletich:

I'm pleased to inform you that your manuscript has been deemed suitable for publication in PLOS ONE. Congratulations! Your manuscript is now with our production department. 

Kind regards, 

on behalf of

Dr. Jennifer Steele 

Academic Editor

PLOS ONE